# Learning and memory function in young people with and without perinatal HIV in England

**Alejandro Arenas-Pinto**[1,2]*, **Ali Judd**[1], **Diane Melvin**[3], **Marthe Le Prevost**[1], **Caroline Foster**[3], **Kate Sturgeon**[1], **Alan Winston**[4], **Lindsay C. Thompson**[1], **Diana M. Gibb**[1], **Hannah Castro**[1], on behalf of the Adolescents and Adults Living with Perinatal HIV (AALPHI) Steering Committee[¶]

1 Medical Research Council Clinical Trials Unit at University College London, London, United Kingdom, 2 Centre for Clinical Research in Infection and Sexual Health, University College London Institute for Global Health, London, United Kingdom, 3 Imperial College Healthcare and Central North West London National Health Service Trust, London, United Kingdom, 4 Imperial College London, London, United Kingdom

¶ Membership of the Adolescents and Adults Living with Perinatal HIV (AALPHI) Steering Committee is listed in the Acknowledgments.
* a.arenas-pinto@ucl.ac.uk

**Data Availability Statement:** The AALPHI data are held at MRCCTU at UCL, which encourages optimal use of data by employing a controlled access approach to data sharing, incorporating a

## Abstract

Learning and memory are important for successful education and career progression. We assess these functions in young people (YP) with perinatal HIV (PHIV) (with or without a previous AIDS-defining illness) and a comparable group of HIV-negative YP. 234 PHIV and 68 HIV-negative YP completed 9 tests; 5 National Institutes of Health (NIH) Toolbox tests (2 executive function, 1 speed of information processing, 2 memory); 2 Hopkins Verbal Learning Test Revised (HVLT-R) (learning (L), delayed recall (R)), and 2 verbal application measures. Z-scores for each test were calculated using normative data and averaged by domain where appropriate. The effect of predictors on test scores in the three domains with the lowest z-scores were analysed using linear regression. 139(59%) and 48(71%) PHIV and HIV-negative YP were female, 202(86%) and 52(76%) Black, and median age was 19 [17, 21] and 18 [16, 21] years respectively. 55(24%) PHIV had a previous Center for Disease Control and Prevention (CDC) class C AIDS-defining diagnosis (PHIV/C). For HVLT-R, there was a trend towards PHIV/C YP having the lowest mean z-scores (L -1.5 (95% CI -1.8,-1.2), R -1.7 (-2.0,-1.4)) followed by PHIV without a CDC C diagnosis (L -1.3 (-1.4,-1.1), R -1.4 (-1.5,-1.2)) and then the HIV-negative group (L -1.0 (-1.3,-0.7), R -1.1 (-1.3,-0.8)); all were greater than 1 SD below the reference mean. The same trend was seen for verbal application measures; however, z-scores were within 1 SD below the reference mean. NIH Toolbox tests were similar for all groups. In multivariable analyses PHIV/C and Black ethnicity predicted lower HVLT-R scores. Black ethnicity also predicted lower executive function scores, however each year increase in age predicted higher scores. In conclusion, cognitive performance in verbal learning and recall fell below population normative scores, and was more pronounced in PHIV/C, supporting wider findings that earlier antiretroviral therapy initiation, before the occurrence of AIDS-defining conditions, may protect aspects of cognitive development.

transparent and robust system to review requests and provide secure data access consistent with the relevant ethics committee approvals. The rationale for this approach has been published (doi: 10.1186/s13063-015-0604-6). Ethics committee approval for use of AALPHI data is restricted to specific approved protocols allowing a controlled access approach. All requests for data are considered and can be initiated by contacting mrcctu.ctuenquiries@ucl.ac.uk.

**Funding:** Funding: The Monument Trust and PENTA Foundation. This work was supported by the Medical Research Council programme grant MC_UU_12023/26 awarded to the MRC Clinical Trials Unit. The funding sources had no role in the study design, collection, analysis and interpretation of data, writing of the report or decision to submit the paper for publication.

**Competing interests:** The authors have declared that no competing interests exist.

## Introduction

There are over two million children and adolescents living with HIV worldwide, the majority of whom acquired it perinatally [1]. Due to the availability of effective antiretroviral therapy (ART), young people living with perinatally acquired HIV (PHIV) are surviving to adulthood, but may have comorbidities resulting from lifelong HIV impacting brain development and neurocognitive functioning [2]. This is important, as cognitive abilities are key contributors to school performance and employability later in life. In the general child population, cognitive processes of executive function predict both reading comprehension and maths outcomes [3], and in PHIV young people, previous studies have shown that prospective and retrospective memory and executive function are key predictors of academic achievements [4].

Evidence from studies assessing neurocognitive status as well as results from neuroimaging in PHIV suggests that early HIV disease progression is associated with CNS impacts which endure into adolescence [5]. Many young people with PHIV who are alive today did not start HIV treatment in early life, under "treat all" approaches, and so potentially missed out on the benefits of this approach for cognitive outcomes [6]. A recent systematic review and meta-analysis found that working memory and executive function were the domains most affected by perinatal HIV infection, with processing speed also being affected [2].

We have previously reported similar performance in neuropsychological testing in young people living with HIV, without a prior history of AIDS-defining complications, and a comparable group of HIV negative (HIV-) young people (YP) within the Adolescents and Adults Living with Perinatal HIV (AALPHI) cohort study [7]. Of note, PHIV with a past history of AIDS-defining conditions showed poorer performance in all cognitive function domains explored. In this paper we report on further cognitive testing performance of young people with PHIV (with or without a previous AIDS-defining illness) and a comparable group of HIV- young people in function domains considered particularly relevant to school progression and transition to adulthood. We explored six function domains with particular focus on executive function, verbal learning and memory because these cognitive functions have been associated with educational and employment outcomes. We also examined predictors of cognitive performance in the three domains with the lowest scores, and additional analyses restricted to participants with PHIV were conducted to investigate the effect of HIV-related factors. For this analysis, we included neuropsychological tests widely used in similar population groups with the aim of generating results easily interpretable in the context of previous studies.

## Methods

The AALPHI cohort was a prospective study evaluating the impact of HIV infection and ART exposure on young people with PHIV and comparing them to HIV- young people. PHIV acquired the infection perinatally whereas HIV- YP were affected by HIV (mainly siblings of PHIV participants or a young person with a parent living with HIV), thus providing a comparison group which had similar background environmental and psychosocial factors related to HIV exposure. Participants were enrolled from HIV clinics and community service locations in England and undertook first interviews between 2013 and 2015, and second interviews between 2015 and 2017. Detailed methods have been described previously [7]. All participants with PHIV had known their HIV status for at least six months, were aged 12–21 years, and were all included in the national UK and Ireland Collaborative HIV Paediatric Study (CHIPS) [8]. HIV- young people were aged 13–23 years and had either a sibling or parent living with HIV. Young people who had been living in the UK for less than six months or were unable to speak or understand English were excluded from the study. Participants provided written informed consent, except where they lacked the capacity, in which case parents or guardians

provided written informed consent and the young person provided written assent. Young people younger than 18 years were allowed to consent to participation in the study themselves if they were deemed by the study research nurses as having the capacity to consent. Full ethical approval was obtained from Leicester Research Ethics Committee.

At the participant's second two-hour face to face interview by research nurses, cognitive assessments (consisting of nine tests exploring six function domains) were carried out. Training was provided on all the cognitive assessments by the study psychologist. Five tests were administered via the computerised National Institutes of Health (NIH) Toolbox® cognition battery (Executive function: flanker inhibitory control and attention test and dimensional change card sort test, Speed of Information Processing: pattern comparison test, and Memory: list sorting and picture sequence test) [9]. Four tests were paper-based, including 2 Hopkins Verbal Learning Test-Revised™ (HVLT-R™) tests for verbal learning (sum of three trials) and verbal delayed recall [10], and 2 verbal application tests: Weschler Individual Achievement Test® Second UK Edition (WIAT®-II^UK) (a word reading task) [11] and British Picture Vocabulary Scale Third Edition (BPVS III) (a word comprehension task) [12].

Data were analysed using STATA version 15.1 (Stata Corp, College Station, Texas, USA). Scoring of all tests followed manufacturer recommendations. For each test, z-scores were calculated using normative data (NIH Toolbox® and HVLT-R™ normative data were adjusted for age, sex, ethnicity and years of education [13, 14], and WIAT®-II^UK and BPVS normative data adjusted for age). Tests designed and validated to assess cognitive function in a domain were grouped and Cronbach α was used as a measure of internal consistency (the lower bound set at 0.6). For domains with more than 1 test, z-scores were averaged to give mean z-scores for each domain. For each domain, mean z-scores were compared across HIV-, PHIV without a CDC clinical classification class C past history of an AIDS-defining illness [15, 16] (PHIV/no C) and PHIV with a CDC C diagnosis (PHIV/C) groups using ANOVA tests. The normality of the mean z-scores was assessed using visual plots and Sharpio-Wilk tests. Characteristics of participants by HIV and CDC status were compared using Chi squared or Fisher's exact tests for proportions (as appropriate given the number of cells with an expected value of five or greater), and Wilcoxon rank sum (comparing two groups) or Kruskal-Wallis (comparing three groups) tests for medians. Results are presented for non-missing values; missing values were less than 10% of study participants unless specified. Ethnicity was self-reported by study participants in the following categories (used by the UK government): White (British, Irish, other), Asian or Asian British (Indian, Pakistani, Bangladeshi, Chinese, other), Black or Black British (Caribbean, African, other), Mixed (White and Black Caribbean, White and Asian, White and Black African, other), Other, Prefer not to say. Ethnicity was analysed as Black (Black or Black British) or non-Black (all other categories) due to small numbers.

The effect of potential predictors on test scores in the three domains with the lowest mean z-scores overall were analysed using linear regression. Factors considered a priori by the AAPLPHI Steering Committee to be associated with test scores were HIV status and CDC disease stage (HIV-, PHIV/no C, PHIV/C), sex, age, ethnicity and country of birth (within vs. outside the UK/Ireland). Other variables considered to be important as potential predictors are shown in Table 1 (denoted with a * asterisk). An additional analysis for only participants with PHIV was also conducted to investigate the effect of HIV-related factors on cognitive performance (HIV-related indicators considered as predictors are shown in Table 2 with a * asterisk; pre-AALPHI clinical data were collected through the Collaborative HIV Paediatric Study). Variables with a P-value <0.15 in univariable analyses were considered in multivariable analyses using backwards selection, as well as the a priori variables, and a P-value <0.05 was considered statistically significant.

**Table 1. Characteristics of participants by HIV and CDC status.**

| | HIV- | PHIV | | | P-value | | |
|---|---|---|---|---|---|---|---|
| | HIV- (n = 68) | No CDC C (n = 179) | CDC C (n = 55) | Total PHIV (n = 234) | HIV- vs PHIV | HIV- vs PHIV/no C vs PHIV/C | PHIV/no C vs PHIV/C |
| *Sociodemographics:* | | | | | | | |
| **Male sex, No. (%)** | 20 (29%) | 68 (38%) | 27 (49%) | 95 (41%) | 0.094 | 0.082 | 0.143 |
| **Age at interview 2, No. (%)** | | | | | | | |
| ≤15 years | 14 (21%) | 22 (12%) | 9 (16%) | 31 (13%) | | | |
| 16–18 years | 26 (38%) | 76 (43%) | 23 (42%) | 99 (42%) | 0.325 | 0.586 | 0.725 |
| ≥19 years | 28 (41%) | 81 (45%) | 23 (42%) | 104 (45%) | | | |
| Median [IQR] | 18 [16,21] | 19 [17,21] | 18 [17, 21] | 19 [17, 21] | 0.453 | 0.600 | 0.474 |
| **Years between interviews, median [IQR]** | 1.7 [1.3,2.4] | 2.0 [1.4,2.6] | 1.8 [1.4, 2.7] | 2.0 [1.4, 2.6] | 0.082 | 0.220 | 0.945 |
| **Ethnicity (Black), No. (%)** | 52 (76%) | 154 (86%) | 48 (87%) | 202 (86%) | 0.050 | 0.144 | 0.815 |
| **Born outside UK/Ireland, No. (%)** | 30 (44%) | 112 (63%) | 24 (44%) | 136 (58%) | 0.040 | 0.006 | 0.013 |
| *Psychosocial:* | | | | | | | |
| **Death of one/both parents\*, No. (%)** | 16 (26%) | 73 (44%) | 19 (36%) | 92 (42%) | 0.021 | 0.039 | 0.297 |
| **Living situation\*, No. (%)** | | | | | | | |
| Family own or rent property | 23 (34%) | 66 (37%) | 22 (40%) | 88 (38%) | | | |
| Housing association/council property | 36 (53%) | 86 (48%) | 29 (53%) | 115 (49%) | 0.834 | 0.669 | 0.365 |
| Other | 9 (13%) | 26 (15%) | 4 (7%) | 30 (13%) | | | |
| **Occupation\*, No. (%)** | | | | | | | |
| Education | 51 (75%) | 132 (74%) | 43 (78%) | 175 (75%) | | | |
| Employment | 11 (16%) | 34 (19%) | 7 (13%) | 41 (18%) | 0.895 | 0.808 | 0.502 |
| Not in education, training or employment | 6 (9%) | 12 (7%) | 5 (9%) | 17 (7%) | | | |
| **Years of education, median [IQR]** | 11 [10,14] | 11 [10, 14] | 11 [10, 14] | 11 [10, 14] | 0.502 | 0.742 | 0.664 |
| **Highest level of parent/carer education\* [a], No. (%)** | | | | | | | |
| Did not complete high school | 4 (7%) | 8 (5%) | 4 (9%) | 12 (6%) | | | |
| High school graduate | 25 (44%) | 59 (41%) | 20 (47%) | 79 (42%) | 0.933 | 0.770 | 0.429 |
| University graduate | 28 (49%) | 79 (54%) | 19 (44%) | 98 (52%) | | | |
| **Ever excluded from school\*, No. (%)** | 17 (25%) | 37 (21%) | 13 (24%) | 50 (22%) | 0.583 | 0.804 | 0.711 |
| *Environmental:* | | | | | | | |
| **Ever fostered/adopted\* [b], No. (%)** | 1 (2%) | 14 (13%) | 4 (12%) | 18 (13%) | 0.049 | 0.118 | 1.000 |
| **Number of adult carers\* [c] (excluding parents)** | | | | | | | |
| Median [IQR] | 0 [0, 1] | 0 [0, 1] | 0 [0, 1] | 0 [0, 1] | 0.862 | 0.472 | 0.232 |
| **Main language spoken at home\* [d], No. (%)** | | | | | | | |
| English only | 30 (44%) | 92 (52%) | 27 (49%) | 119 (51%) | | | |
| English and another equally | 34 (50%) | 82 (46%) | 25 (45%) | 107 (46%) | 0.388 | 0.535 | 0.473 |
| Languages other than English | 4 (6%) | 4 (2%) | 3 (5%) | 7 (3%) | | | |
| **Young carer\*, No. (%)** | 21 (31%) | 16 (9%) | 7 (13%) | 23 (10%) | <0.001 | <0.001 | 0.424 |
| *Lifestyle:* | | | | | | | |
| **Ever smoked\*, No. (%)** | 27 (40%) | 72 (41%) | 24 (44%) | 96 (42%) | 0.854 | 0.871 | 0.623 |
| **Ever consumed alcohol\*, No. (%)** | 44 (65%) | 117 (65%) | 27 (50%) | 144 (62%) | 0.664 | 0.113 | 0.042 |
| **Waist to hip ratio\*, median [IQR]** | 0.9 [0.8,0.9] | 0.9 [0.8,0.9] | 0.9 [0.8, 0.9] | 0.9 [0.8, 0.9] | 0.397 | 0.203 | 0.122 |
| **Waist to hip ratio z-score [e], median [IQR]** | 1.2 [0.2, 2.1] | 1.2 [0.3,2.0] | 1.4 [0.6,2.4] | 1.3 [0.4,2.1] | 0.665 | 0.544 | 0.300 |
| *Mental health:* | | | | | | | |

*(Continued)*

**Table 1.** (Continued)

| | HIV- | PHIV | | | P-value | | |
|---|---|---|---|---|---|---|---|
| | HIV-<br>(n = 68) | No CDC C<br>(n = 179) | CDC C<br>(n = 55) | Total PHIV<br>(n = 234) | HIV- vs<br>PHIV | HIV- vs PHIV/no C vs<br>PHIV/C | PHIV/no C vs<br>PHIV/C |
| EQ-5D-5L Anxiety/Depression[*][f], No. (%) | | | | | | | |
| No anxiety or depression | 41 (61%) | 109 (62%) | 30 (57%) | 139 (61%) | | | |
| Slightly/moderately anxious/depressed | 21 (31%) | 56 (32%) | 18 (34%) | 74 (32%) | 0.983 | 0.930 | 0.657 |
| Severely/extremely anxious/depressed | 5 (8%) | 11 (6%) | 5 (9%) | 16 (7%) | | | |

Abbreviations: HIV-, HIV negative; IQR, interquartile range; PHIV, perinatal HIV; PHIV/C, PHIV with a CDC C diagnosis; PHIV/no C, PHIV without a CDC C diagnosis

[*] variables considered as predictors in multivariable models (as well as HIV status and CDC disease stage, sex, age, ethnicity and country of birth as a priori variables)

[a] unknown for 11 HIV-, 33 PHIV/no C, 12 PHIV/C;

[b] unknown for 25 HIV-, 73 PHIV/no C, 21 PHIV/C (question was introduced part way through interview 2);

[c] number of adults taking responsibility for and living with the young person during childhood excluding parents;

[d] at the time of interview 1;

[e] calculated with reference to healthy uninfected children, adjusted for age and sex [17];

[f] anxiety/depression dimension of the EQ-5D-5L instrument was used to measure anxiety and depression [18]

## Results

A total of 234 PHIV and 68 HIV- young people completed cognitive testing at interview 2 and 55 (24%) of PHIV young people had a CDC C diagnosis. There were more females than males in each group, most were Black and around half were born outside of the UK/Ireland (Table 1). The median age was 19 (IQR 17,21) and 18 (16,21) years (p = 0.453) for the PHIV and HIV- groups respectively and the majority of young people were still in education in each group. For 42% of PHIV and 26% of HIV- YP one or both parents had died, more PHIV young people had been fostered or adopted (13% PHIV, 2% HIV-) and more HIV- young people were young carers (10% PHIV, 31% HIV-). There were no other statistically significant differences between the groups in psychosocial, environmental or mental health characteristics (Table 1). In the PHIV group, those with a CDC C diagnosis were more likely to have presented to HIV care and initiated ART at a younger age and in earlier calendar years, than those without a CDC C diagnosis (Table 2). Most had started ART with 3 or 4 drugs and a regimen containing a non-nucleoside reverse transcriptase inhibitor (NNRTI) or boosted protease inhibitor (PI).

Cognitive function mean test and z-scores by domain, HIV and CDC status are shown in S1 Table. Mean z-scores and 95% confidence intervals were below reference means in nearly all HIV/CDC groups and domains, suggesting lower cognitive performance compared with normative data (Fig 1). For verbal learning (L) and verbal delayed recall (R), there was a trend towards PHIV/C young people having the lowest mean z-scores (L -1.5 (95% CI -1.8,-1.2), R -1.7 (-2.0,-1.4)) followed by PHIV/no C (L -1.3 (-1.4,-1.1), R -1.4 (-1.5,-1.2)) and then HIV- young people (L -1.0 (-1.3,-0.7), R -1.1 (-1.3,-0.8)) (L p = 0.026, R p = 0.004), and all were greater than minus one standard deviation below the reference mean. A similar trend was seen for verbal application measures (PHIV/C -0.5 (-0.7,-0.3), PHIV/no C -0.2 (-0.4,-0.1), HIV- -0.1 (-0.3,0.1), p = 0.034), however they were within one standard deviation below the reference mean for all groups, suggesting that these skills were less compromised than verbal

**Table 2. HIV-related characteristics of PHIV participants by CDC status.**

| | PHIV | | | P-value |
|---|---|---|---|---|
| | **No CDC C (n = 179)** | **CDC C (n = 55)** | **Total (n = 234)** | **PHIV/no C vs PHIV/C** |
| **Age at first presentation\*, No. (%)** | | | | |
| Birth | 14 (8%) | 4 (7%) | 18 (8%) | |
| <1 year | 16 (9%) 52 (25%) | 16 (30%) | 32 (14%) | |
| 1–4 years | 49 (27%) | 14 (26%) | 63 (27%) | 0.003 |
| 5–9 years | 58 (32%) | 13 (24%) | 71 (30%) | |
| ≥10 years | 42 (23%) | 7 (13%) | 49 (21%) | |
| Median [IQR] | 6.0 [2.1,9.4] | 2.2 [0.3,7.1] | 5.2 [1.5,9.2] | 0.005 |
| **Year of first presentation\*, No. (%)** | | | | |
| Up to 1996 | 28 (16%) | 9 (17%) | 37 (16%) | |
| 1997–2000 | 32 (18%) | 20 (37%) | 52 (22%) | 0.008 |
| 2001 onwards | 119 (66%) | 25 (46%) | 144 (62%) | |
| **ART status at interview\*, No. (%)** | | | | |
| Never started ART | 12 (7%) | 0 (0%) | 12 (5%) | |
| On ART | 155 (86%) | 52 (95%) | 207 (89%) | 0.135 |
| Off ART (previous ART exposure) | 12 (7%) | 3 (5%) | 15 (6%) | |
| **Age at initiation of ART\*, median[IQR] (on or off ART)** | 9.2 [4.8,12.8] | 3.6 [0.5,8.5] | 7.5 [3.4,12.2] | <0.001 |
| **Year initiated ART\*, No. (%) (on or off ART)** | | | | |
| Up to 1996 | 11 (7%) | 4 (7%) | 15 (7%) | |
| 1997–2000 | 25 (15%) | 25 (46%) | 50 (22%) | <0.001 |
| 2001 onwards | 131 (78%) | 26 (47%) | 157 (71%) | |
| **No. of drugs at ART initiation\*, No. (%) (on or off ART)** | | | | |
| 1 or 2 | 19 (11%) | 7 (13%) | 26 (12%) | 0.787 |
| 3 or 4 | 148 (89%) | 48 (87%) | 196 (88%) | |
| **Class at ART initiation\*, No. (%) (on or off ART)** | | | | |
| NNRTI based | 107 (64%) | 31 (56%) | 138 (62%) | |
| Boosted PI based | 28 (17%) | 7 (13%) | 35 (16%) | 0.182 |
| Unboosted PI based/NRTI only | 32 (19%) | 17 (31%) | 49 (22%) | |
| **Taking efavirenz at interview\*, No. (%) (on ART only)** | 37 (24%) | 9 (17%) | 46 (22%) | 0.325 |
| **Viral load <50 copies/ml at interview\* [a], No. (%) (on ART only)** | 101 (71%) | 32 (64%) | 133 (71%) | 0.221 |
| **Median[IQR] cumulative years viral load <400 copies/ml\*** | 6.9 [3.9,10.0] | 6.9 [4.0,11.9] | 6.9 [3.9,10.2] | 0.348 |
| **Median[IQR] cumulative years viral load <50 copies/ml\* <50 copies/ml\*\*** | 6.0 [2.7,9.1] | 6.2 [2.9,10.4] | 6.1 [2.8,9.3] | 0.524 |
| **CD4 nadir (cells/mm³)\*, medium [IQR]** | 234 [168,380] | 166 [17,330] | 234 [131,359] | 0.003 |
| **CD4 at interview\* [a,b] (cells/mm³), medium [IQR]** | 555 [402,784] | 641 [485,776] | 581 [407,782] | 0.260 |

Abbreviations: IQR, interquartile range; PHIV, perinatal HIV; PHIV/C, PHIV with a CDC C diagnosis; PHIV/no C, PHIV without a CDC C diagnosis; NNRTI, non-nucleoside reverse transcriptase inhibitor; PI, protease inhibitor; NRTI, nucleoside reverse transcriptase inhibitor.

\* additional variables considered as predictors in multivariable models in the PHIV group only.

[a] within 6 months before or after interview;

[b] unknown for 33 PHIV/no C, 6 PHIV/C

learning and verbal delayed recall. Executive function, speed of information processing and memory tests z-scores were similar for all three groups.

Multivariable analyses looked at predictors of improved verbal learning (L), verbal delayed recall (R), and executive function (Flanker inhibitory control and attention (F), and dimensional change card sort (D)) scores (S2 Table). Both before and after adjustment for other

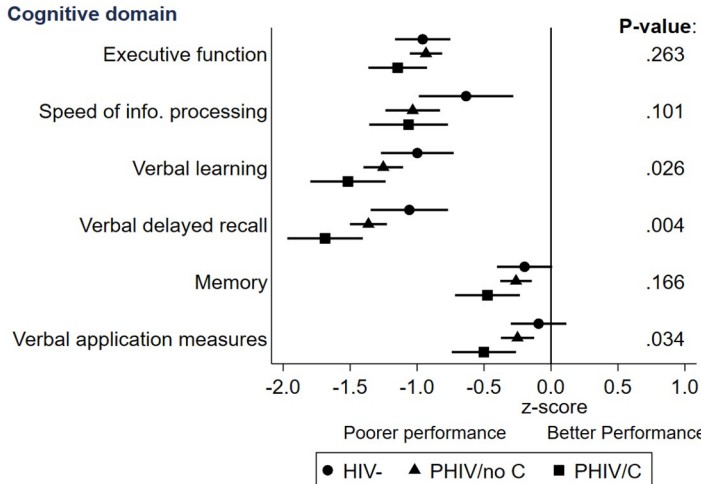

**Fig 1. Cognitive performance by domain and HIV/CDC status.** Data points are means with 95% confidence intervals. P-values compare the three groups (HIV, PHIV/no C, PHIV/C) using analysis of variance. Abbreviations: CDC, Centers for Disease Control and Prevention, HIV-, HIV negative; PHIV, perinatal HIV; PHIV/C, PHIV with a CDC C diagnosis; PHIV/no C, PHIV without a CDC C diagnosis.

factors, verbal learning and verbal delayed recall scores were lower in PHIV/C young people (multivariable coefficient L -2.55 (95% confidence interval -4.25, -0.85); R -1.18 (-2.10, -0.25), p = 0.040) and young people of Black ethnicity (R -2.59 (-4.11, -1.06); L -1.53 (-2.34, -0.73), p<0.001). Executive function scores were also lower in Black ethnicity young people (F -0.30 (-0.58, -0.01), p = 0.040; D -0.35 (-0.67, -0.04), p = 0.029) and improved with each year increase in age (F 0.04 (0.00, 0.08), p = 0.033; D 0.09 (0.04, 0.13), p<0.001). There were also some factors which were only predictors for one of the four tests in multivariable analysis; higher waist to hip ratio was associated with lower verbal delayed recall scores (-4.76 (-8.63, -0.88) per 0.1 increase in waist to hip ratio, p = 0.016), ever having smoked was associated with higher flanker inhibitory control and attention scores (0.22 (0.01, 0.43), p = 0.039), and loss of a parent was associated with lower dimensional change card sort scores (-0.32 (-0.56, -0.09), p = 0.007) as well as being a young carer (-0.39 (-0.72, -0.05), p = 0.024).

A separate model for the PHIV group found similar results to the overall model in terms of associations with CDC status, Black ethnicity and age (S3 Table). In multivariate analysis, having a supressed viral load <50 copies/ml at the time of the interview was associated with higher verbal learning scores (multivariable coefficient 2.25 (0.90, 3.61) p = 0.001) and starting ART with a regimen containing a boosted PI was associated with higher verbal delayed recall scores (1.48 (0.51, 2.45) p = 0.011).

## Discussion

In this well characterised cohort of young people with perinatally acquired HIV infection and a comparable group of HIV- young people, we found similar levels of performance across the three HIV/CDC groups and for most of the cognitive domains. However, performance was lowest in the verbal learning and delayed recall domains, and the lowest performance was consistently seen in PHIV/C young people. The scores of PHIV/C young people on verbal application measures were also lower than in the other two groups, but the mean z-score for all groups were within 1 SD below the normative mean.

Poorer cognitive performance in PHIV/C young people has been observed in other studies. Malee et al found PHIV with past AIDS-defining diagnoses had lower performance across

multiple executive function tasks than PHIV/no C and perinatally HIV-exposed uninfected youth (PHEU) [19]. In another study, 45 PHIV/C had significantly lower visual recognition memory compared with 85 PHEU [20]. A study in south-east Asia reported PHIV performing significantly worse on visual memory tests when compared with PHEU at baseline, but found no difference between groups in the trajectory of working memory measures over the three year follow-up period [21].

Consistent with previous reports, performance of the entire cohort in most cognitive domains was lower than normative population means and may be lower than we have reported, as those with severe cognitive impairment may not have been referred to the study. This suggests that performance in cognitive tasks may be tempered by factors and influences not related to HIV disease and highlights the importance of having an appropriate comparison group [4, 19, 22]. However, a longitudinal analysis comparing cognitive testing performance in children with perinatally acquired HIV and HIV-exposed but uninfected children in Zambia, recently reported that socioeconomic factors impacted testing performance in both groups, but with a more marked effect in the PHIV group [23]. Another study in South Africa showed uninfected children exposed to maternal HIV infection had language delays at 2 years of age, so further work is needed to understand developmental outcomes of this population [24].

Executive function scores improved with increasing age but age was not associated with improved verbal learning or delayed recall scores. The relative contribution of specific cognitive abilities may vary according to age, and previous studies have shown that working memory and concrete and visual reasoning dominate in primary school whereas language and more abstract reasoning appear to do so in secondary school [25, 26]. Abdominal obesity was associated with poorer verbal delayed recall, but not other cognitive function domains. Abdominal obesity has been associated with poorer cognitive performance in adults living with HIV [27] and in children in the general population [28, 29]. However, to our knowledge no data have been published on this association in young people living with HIV and further research is needed.

In contrast with findings at the first interview on this cohort, which whilst used different cognitive assessments measured similar domains [7] (tests at the second interview were easier to administrate and had more appropriate normative data for our population), two years later we observed no difference in performance between PHIV with and without CDC C diagnoses in speed of information processing, memory and executive functioning. This may well reflect ongoing brain development during adolescence [30] in the absence of significant further HIV-related insults such as opportunistic infections, and is consistent with previous studies in which cognition seems to improve [21, 29] or to remain stable over time in the short term [19]. We detected an effect of ethnicity on cognition in this analysis, as we did in our previous paper. Although the NIH Toolbox and HVLT-R normative data are adjusted for ethnicity, their normative sample was around 60% and 58% White, and 15% and 42% African American respectively, and this may contribute to the difference we found [13, 14].

Our study has a number of limitations. Firstly, the tools we used to investigate cognitive performance differed at the different study visits and therefore we only have cross-sectional data on the associations between these measures and HIV status. However, if practice effect was a factor to be considered, using different tests to assess performance on a given cognitive domain may be advantageous. Secondly, the study was unable to ascertain any effect of different antiretroviral drug or drug class on cognition.

In conclusion, our study showed similar performance in cognitive testing between PHIV and HIV- young people in most function domains. However, performance in verbal learning and recall fell below population normative scores, and was more pronounced in PHIV/C,

supporting wider findings that early initiation of ART, before the occurrence of AIDS-defining conditions, may protect aspects of cognitive development.

## Supporting information

**S1 Table. Cognitive function test and z-scores by domain, HIV and CDC status.** Mean scores were compared across the three groups (HIV-, PHIV/no C, PHIV/C) using ANOVA tests. Abbreviations: HIV-, HIV negative; PHIV, perinatal HIV; PHIV/C, PHIV with a CDC C diagnosis; PHIV/no C, PHIV without a CDC C diagnosis.
(DOCX)

**S2 Table. Predictors of improved verbal learning, verbal delayed recall and executive function scores.** All a priori variables, as well as those with univariable P-value <0.15 and multivariable P-value <0.05 are presented. Abbreviations: CI, confidence interval, HIV-, HIV negative; PHIV, perinatal HIV; PHIV/C, PHIV with a CDC C diagnosis; PHIV/no C, PHIV without a CDC C diagnosis.
(DOCX)

**S3 Table. Predictors of improved verbal learning, verbal delayed recall and executive function scores for PHIV participants.** All a priori variables, as well as those with univariable P-value <0.15 and multivariable P-value <0.05 are presented. Abbreviations: CI, confidence interval, HIV-, HIV negative; PHIV, perinatal HIV; PHIV/C, PHIV with a CDC C diagnosis; PHIV/no C, PHIV without a CDC C diagnosis; NNRTI, non-nucleoside reverse transcriptase inhibitor; PI, protease inhibitor; NRTI, nucleoside reverse transcriptase inhibitor. [a] within 6 months before or after interview.
(DOCX)

## Acknowledgments

We thank all young people, parents and staff from all the clinics and voluntary services participating in AALPHI.

Membership of the Adolescents and Adults Living with Perinatal HIV (AALPHI) Steering Committee is as follows:

Project team: S. Brice, H. Castro, A. Judd, M. Le Prevost, A. Mudd, A. Nunn, K. Rowson, K. Sturgeon, L.C. Thompson.

Investigators: M. Conway, K. Doerholt, D. Dunn, C. Foster, D.M. Gibb, A. Judd (PI), S. Kinloch, N. Klein, H. Lyall, D. Melvin, K. Prime, T. Rhodes, C. Sabin, M. Sharland, C. Thorne, P. Tookey.

MRC CTU Data Services: C. Diaz Montana, K. Fairbrother, M. Rauchenberger, N. Tappenden, S. Townsend.

Neurocognitive subgroup: A. Arenas-Pinto, H. Castro, C. Foster, A. Judd, M. Le Prevost, D. Melvin, A. Winston.

Steering Committee chairs: D. M. Gibb, D. Mercey (2012–2015), C. Foster (2016-).

Patient and public involvement: Children's HIV Association Youth Committee

NHS clinics (named alphabetically): LONDON: Chelsea and Westminster NHS Foundation Trust, F. Boag, P. Seery; Great Ormond Street Hospital NHS Foundation Trust, M. Clapson, V. Noveli; Guys and St Thomas' NHS Foundation Trust, A. Callahan, E. Menson; Imperial College Healthcare NHS Trust, C. Foster, A. Walley; King's College Hospital NHS Foundation Trust, E. Cheserem, E. Hamlyn; Mortimer Market Centre, Central and North West London NHS Foundation Trust, R. Gilson, T. Peake; Newham University Hospital, S. Liebeschuetz, R. O'Connell; North Middlesex University Hospital NHS Trust, J. Daniels, A. Waters; Royal Free

London NHS Foundation Trust, T. Fernandez, S. Kinloch de Loes; St George's University Hospitals NHS Foundation Trust, S. Donaghy, K. Prime. REST OF ENGLAND: Alder Hey Children's NHS Foundation Trust, S. Paulus, A. Riordan; Birmingham Heartlands, Heart of England NHS Foundation Trust J. Daglish, C. Robertson; Bristol Royal Infirmary, University Hospitals Bristol NHS Foundation Trust, J. Bernatonlene, L. Hutchinson, University Hospitals Bristol NHS Foundation Trust, M. Gompel, L. Jennings; Leeds Teaching Hospitals NHS Trust, M. Dowie, S. O'Riordan; University Hospitals of Leicester NHS Trust, W. Ausalut, S. Bandi; North Manchester General Hospital, Pennine Acute Hospitals NHS Trust, P. McMaster, K. Rowson; Royal Liverpool and Broadgreen University Hospitals NHS Trust, M. Chaponda, S Paulus.

Voluntary services (named alphabetically): Blue Sky Trust, C. Dufton, B. Oliver; Body and Soul, A. Ash, J. Marsh; Faith in People, I. Clowes, M. Overton; Positively UK, M. Kiwanuka, A. Namiba; Positive Parenting & Children, N. Bengtsson, B. Chipalo.

Lead contact for the AALPHI Steering Committee: Prof Ali Judd, a.judd@ucl.ac.uk

## Author Contributions

**Conceptualization:** Alejandro Arenas-Pinto, Ali Judd, Diane Melvin, Marthe Le Prevost, Caroline Foster, Kate Sturgeon, Alan Winston, Lindsay C. Thompson, Diana M. Gibb, Hannah Castro.

**Data curation:** Marthe Le Prevost, Kate Sturgeon.

**Formal analysis:** Hannah Castro.

**Methodology:** Ali Judd.

**Writing – original draft:** Alejandro Arenas-Pinto, Hannah Castro.

**Writing – review & editing:** Alejandro Arenas-Pinto, Ali Judd, Diane Melvin, Marthe Le Prevost, Caroline Foster, Kate Sturgeon, Alan Winston, Lindsay C. Thompson, Diana M. Gibb.

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
