## [Decision Letter · Decision Letter 0]

6 Dec 2021

PONE-D-21-16526

Learning and memory function in young people with and without perinatal HIV in England

PLOS ONE

Dear Dr. Arenas-Pinto,

Thank you for submitting your manuscript to PLOS ONE. After careful consideration, we feel that it has merit but does not fully meet PLOS ONE’s publication criteria as it currently stands. Therefore, we invite you to submit a revised version of the manuscript that addresses the points raised during the review process.. The reviewers and I agree that this work is of importance and that it should be published but both reviewers have highlighted areas where the paper can be strengthened to have a greater impact.  In particular, without restating the points in the reviews - the analyses section needs reconsideration and cleaning up. All data should be provided so that readers can make their own determination and the combining of test scores should be given more thought.  This was seen as a minor revision by Reviewer one and as a major revision by Reviewer 2 but both believed that this aspect of the study could be much improved and that a more sophisticated and nuanced approach to test interpretation was needed.  Both reviewers also questioned whether the introduction adequately positioned the study within the literature and reviewer 2 wanted an indication of what this study adds to your previous reporting of the same data set.  It will be important for your revision to address these concerns and by doing so I am confident you will greatly improve your manuscript. We were very fortunate to have such thoughtful reviewers and I appreciate the time that they spent lending their expertise to help strengthen your paper. Please ensure that your revision addresses their comments. I look forward to seeing a resubmission of this work.

We look forward to receiving your revised manuscript.

Kind regards,

Lori Buchanan, Ph.D.

Academic Editor

PLOS ONE

Journal Requirements:

2. Please note that in order to use the direct billing option the corresponding author must be affiliated with the chosen institute. Please either amend your manuscript to change the affiliation or corresponding author, or email us at plosone@plos.org with a request to remove this option.

3. One of the noted authors is a group or consortium "Adolescents and Adults Living with Perinatal HIV (AALPHI) Steering Committee". In addition to naming the author group, please list the individual authors and affiliations within this group in the acknowledgments section of your manuscript. Please also indicate clearly a lead author for this group along with a contact email address.

Reviewers' comments:

Reviewer's Responses to Questions

**Comments to the Author**

1. Is the manuscript technically sound, and do the data support the conclusions?

Reviewer #1: Partly

Reviewer #2: Yes

2. Has the statistical analysis been performed appropriately and rigorously? 

Reviewer #1: No

Reviewer #2: I Don't Know

3. Have the authors made all data underlying the findings in their manuscript fully available?

Reviewer #1: No

Reviewer #2: No

4. Is the manuscript presented in an intelligible fashion and written in standard English?

Reviewer #1: Yes

Reviewer #2: Yes

5. Review Comments to the Author

Reviewer #1: Thank you for the opportunity to review this manuscript. The authors sought to compare the cognitive functioning of three groups: people with PHIV and CDC C diagnosis, people with PHIV and no CDC C diagnosis, and HIV- individuals with a sibling/parent who has HIV.

This is a clinically relevant and important issue for a wide variety of clinicians. Understanding cognitive functioning in these groups has important implications for clinical service allocation, school/work accommodations, and other support. The HIV- control group is a strength of this study. However, the manuscript would benefit from a more thorough review of the literature; clarification of methods; and a more nuanced discussion of cognitive test findings. Please find my suggestions below.

Major Revisions

1. The manuscript requires better alignment of aims, methods, and results. For example, the aim in the abstract states, “we assess these functions in young people (YP) with perinatal HIV (PHIV) and a comparable group of HIV-negative (HIV-) YP.” However, the methods described three groups. Additionally, the aim is to compare groups on cognitive functioning, but a regression model is run to assess predictors of cognitive functioning. No objective or hypothesis for the regression analysis is reported, nor are the results of the regression reported in the abstract.

2. Overall, the findings need to be better contextualized within the broader literature in both the introduction and the discussion. For the introduction, what is already known about the effects of perinatal HIV on cognition in young people? What don’t we know about cognitive functioning in this population? Why is this study important? Moreover, the authors cite a previously published paper from their own working group on cognitive performance in the same cohort. These findings should be reported in the introduction. It is unclear how this study differs from their previously published paper.

3. For domains with more than one test, z scores were averaged to create a mean z score for each domain. This seems appropriate; however, the rationale for combining certain tests needs to be made explicit. From a neuropsychological perspective, some of the combinations may not be appropriate. For example:

a. What is the rationale for combining list sorting and picture sequence from the NIHTB? The list sorting test is a verbal working memory task and picture sequence is a visual episodic memory task. Conceptually, a working memory task might be better defined as an executive functioning task. See https://pubmed.ncbi.nlm.nih.gov/24959983/.

b. The picture memory task shows good convergent validity with the RAVLT (a very common list learning test), which is similar to the HVLT (see https://www.ncbi.nlm.nih.gov/pmc/articles/PMC4254833/). What is the rational for calling the HVLT a learning task and separating it from the other memory tasks? Learning is a component of memory. If the HVLT is impaired this is a component of verbal memory.

4. Although the results for the Verbal Learning PHIV C vs PHIV-C are statistically different, the confidence intervals are completely overlapping. Moreover, the z-scores are very similar (1.3 vs 1.5). This translates to a standard score of 78 vs 81, respectively. Is this really clinically meaningful? These scores may result in very similar behavioral presentation. Effect sizes should be reported to demonstrate magnitude of effects.

5. More clarity is warranted with regard to the aim and methods of the regression analysis:

a. What is the goal of the regression analysis? If part of the primary objective of the paper is to determine what factors might predict poorer cognitive functioning in this group, that should be made explicit in the introduction/aims.

b. IQ/education are important predictors of cognitive functioning. Were these controlled for in the regression to determine what factors predict poor outcomes? If not, why?

c. The authors note that the predictors used in the regression can be found in Table 1. Table 1 has over 20 variables. Are all of these included in the regression? I would think the analysis would be underpowered for that many variables. Please clarify and provide justification for the predictors.

d. An additional analysis with only PHIV was conducted. What is the rationale for this? Was this decided a priori or is this a sensitivity analysis? This needed clarification.

6. Raw score means, SDs, and ranges for all cognitive testing should be presented. This is necessary for interpretation of the results.

Minor Revisions by Section

Abstract

1. The title and Abstract background make it sound like only learning and memory was assessed. Consider updating both.

2. Acronyms are not defined before use. (e.g., CDC C diagnosis, ART)

3. The original aim is to compare PHIV to HIV-PV, but there is a three-group comparison reported in results.

4. Results of the linear regressions are not mentioned.

5. The conclusion made in the abstract that results suggest earlier ART leads to better outcome is not support by the results reported in the abstract. There is no mention of ART in the result section.

Introduction

6. The connection to education in the introduction is a little unclear. This might be an important implication of the work; however, it is not directly related to the study aim.

7. A rationale should be provided for using the HIV-negative control group. I assume this control group accounts for environmental and psychosocial factors related to HIV exposure. This should be made explicit.

8. In line with the above, there is clearly good reason for comparing groups with and without a CDC C diagnosis separately, but the rationale should be made explicit.

9. Why are no a prioiri hypotheses reported?

Method

10. The first sentence needs editing. It reads like the study was evaluating the impact of HIV infection and ART on HIV- young people as it is currently written.

11. Please report any exclusion criteria for the larger study.

12. A lot of space is used to discuss the consent/assent process. This is good/ethical practice, but unnecessarily detailed for a manuscript.

13. Cognitive testing described to have occurred during a face-to-face interview. It may be better described as an interview with cognitive testing.

14. Please define the CDC Clinical classification class C past history of an AIDS defining illness.

Statistical analysis

15. Specify what comparisons are being made using chi square tests/Wilcoxon/KW.

Results

16. Were there any significant differences between your groups on important demographic or clinical factors such as education, SES, psychiatric diagnoses? I believe these comparisons are made in Table 1, but the authors should include a statement in the results and then point readers to the table.

17. Information on variable distribution/assumptions underlying statistical analyses should be reported.

Discussion

18. It would be interesting to discuss in more detail why your control group also performed lower than the normative sample on several tests to contextualize the results. For example, what common factors between all three groups could have impacted cognitive performance on these measures and is this consistent with the literature?

19. Based on what is reported in the discussion, the HIV+ cohort in this study had impairments in cognitive domains that differed from previously published worked. This should be discussed.

20. There is a note to add “blackmore reference” in the discussion.

21. It is unclear why the second limitation is a limitation of the study. Was identifying differential effect of antiretroviral drugs on cognition a goal? Is there a reason to believe that different treatments affect cognition in different ways? If so, this should be made more explicit.

22. A limitation of this study is only using active recall to test memory (i.e., no recognition memory test from HVLT), therefore it is unclear if deficits in delayed memory extend to recognition. If this cohort has impaired learning and recall, but not encoding (i.e., recognition memory is average), this would suggest a frontal executive, as opposed to amnestic mesial temporal profile.

23. The authors conclude that their findings support that earlier ART protects against cognitive development. Was age of ART included in the regression model? Did it predict outcomes? It seems you have the data to test this assumption, but it is not clearly reported anywhere in the manuscript.

Tables

24. Table 1 is very large. I would recommend splitting up into more than one table.

25. Should present results of the linear regressions in a table for transparency.

Reviewer #2: Abstract

1. Background paragraph of abstract emphasizes relevance of learning and memory for successful educational and career progression, but this was not examined in this study. Authors should consider a more relevant background justification that would more directly apply to this work and supporting analyses.

2. “CDC C” diagnosis term in abstract would not be known to a general scientific audience. Consider briefly defining.

Methods

1. The section on the consent process can be shortened/streamlined.

2. It would be helpful for the authors to briefly explain why it would be relevant to compare PHIV/no C and PHIV/C. This is done within the discussion, but it would be beneficial for the reader to understand the reasoning for this comparison from the outset (perhaps within the introduction?)

3. Executive functioning and memory performance can be directly impacted by attention, and attention difficulties are documented in HIV. Was there a reason attention measures were not included in this battery?

Results

1. While it is logical that multiple test scores within a domain would be averaged to present results in a cohesive manner, the authors should make explicit that the scores were sufficiently statistically similar for this to be appropriate. Please include a table summarizing mean performance per group for all individual tests using a common metric (e.g., z scores). Also explain the criteria for averaging across tests within a domain; for example, did the authors ensure that mean test scores within a domain were not statistically different from each other? I imagine that there may have been some variability in performance for within-domain tests. Is it possible that potential effects were attenuated because of variable performance within domains?

Discussion

1. There were some lower cognitive scores amongst participants of Black race found in adjusted multivariable models. It is important that the authors address potential explanations for this within the discussion.

General comment for consideration:

1. The term “Black” generally refers to a race/racial category, as you can have participants who are “Black” but report different ethnic backgrounds. The authors can consider these recently published suggestions for reporting of race or ethnicity: https://jamanetwork.com/journals/jama/fullarticle/2776936

6. PLOS authors have the option to publish the peer review history of their article (what does this mean?). If published, this will include your full peer review and any attached files.

Reviewer #1: No

Reviewer #2: No

---

## [Author Response · Author response to Decision Letter 0]

4 Mar 2022

PONE-D-21-16526

Learning and memory function in young people with and without perinatal HIV in England

PLOS ONE

We thank the journal and reviewers for their time considering our paper. We are delighted with the opportunity of submitting a revised version of our manuscript for your consideration and, following your request, we have addressed the comments made by your editorial office and the reviewers below.

Academic editor comments:

1. The analyses section needs reconsideration and cleaning up. All data should be provided so that readers can make their own determination and the combining of test scores should be given more thought. This was seen as a minor revision by Reviewer one and as a major revision by Reviewer 2 but both believed that this aspect of the study could be much improved and that a more sophisticated and nuanced approach to test interpretation was needed. 

Response: More detail has been added to the methods and results sections, including the addition of supplementary tables detailing the raw data from the individual tests and results of the regression analyses. We have also added more details on the rationale of combining some of the tests into domains.

2. Both reviewers also questioned whether the introduction adequately positioned the study within the literature and reviewer 2 wanted an indication of what this study adds to your previous reporting of the same data set.

Response: We have added more details to the introduction about our previous study looking at cognitive function at interview 1.

Please see our responses to individual comments below.

Journal Requirements:

Response: This has been done.

2. Please note that in order to use the direct billing option the corresponding author must be affiliated with the chosen institute. Please either amend your manuscript to change the affiliation or corresponding author, or email us at plosone@plos.org with a request to remove this option.

Response: The corresponding author is affiliated to two Institutes from University College London, the Institute of Clinical Trials and Methodology (ICTM) and the Institute for Global Health (IGH). The MRC Clinical Trials at UCL is based at ICTM. It is not clear to us where the confusion is, but would be happy to work with you to get this sorted.

3. One of the noted authors is a group or consortium "Adolescents and Adults Living with Perinatal HIV (AALPHI) Steering Committee". In addition to naming the author group, please list the individual authors and affiliations within this group in the acknowledgments section of your manuscript. Please also indicate clearly a lead author for this group along with a contact email address.

Response: The membership and lead contact for the AALPHI steering committee is already stated in the Acknowledgements section. We have added a note to the title page to indicate that membership of the group is provided in the Acknowledgements.

Response: The data have now been included in a Supporting Information file (S3 Table).

Reviewers' comments: 

Reviewer #1: 

Major Revisions

1. The manuscript requires better alignment of aims, methods, and results. For example, the aim in the abstract states, “we assess these functions in young people (YP) with perinatal HIV (PHIV) and a comparable group of HIV-negative (HIV-) YP.” However, the methods described three groups. Additionally, the aim is to compare groups on cognitive functioning, but a regression model is run to assess predictors of cognitive functioning. No objective or hypothesis for the regression analysis is reported, nor are the results of the regression reported in the abstract.

Response: We have amended the abstract and introduction to better reflect the three groups compared in the analysis. We have also included the regression analysis in the abstract and added it as an aim in the introduction.

2. Overall, the findings need to be better contextualized within the broader literature in both the introduction and the discussion. For the introduction, what is already known about the effects of perinatal HIV on cognition in young people? What don’t we know about cognitive functioning in this population? Why is this study important? Moreover, the authors cite a previously published paper from their own working group on cognitive performance in the same cohort. These findings should be reported in the introduction. It is unclear how this study differs from their previously published paper.

Response: As the reviewer mentioned, we have explored cognitive function performance in the AALPHI cohort at baseline. In the initial assessment we found no overall difference in neuropsychological testing performance between young people living with HIV, who have never developed any AIDS-defining complication, and HIV-negative young people. However, performance on tests exploring memory and information processing cognitive domains seem to be poorer in PHIV, particularly those who have experienced AIDS-defining complications in the past. Therefore, the round of neuropsychological assessment we are presenting in this manuscript, focused predominately on learning and memory cognitive function domains because of their implications on educational and employment outcomes.

We have amended the introduction to reflect this. 

3. For domains with more than one test, z scores were averaged to create a mean z score for each domain. This seems appropriate; however, the rationale for combining certain tests needs to be made explicit. From a neuropsychological perspective, some of the combinations may not be appropriate. For example:

a. What is the rationale for combining list sorting and picture sequence from the NIHTB? The list sorting test is a verbal working memory task and picture sequence is a visual episodic memory task. Conceptually, a working memory task might be better defined as an executive functioning task. See https://pubmed.ncbi.nlm.nih.gov/24959983/.

b. The picture memory task shows good convergent validity with the RAVLT (a very common list learning test), which is similar to the HVLT (see https://www.ncbi.nlm.nih.gov/pmc/articles/PMC4254833/). What is the rational for calling the HVLT a learning task and separating it from the other memory tasks? Learning is a component of memory. If the HVLT is impaired this is a component of verbal memory.

Response: We do agree that the findings do not clarify the specific neurocognitive processes most affected by the presence of HIV in the developing brain. The choice of tests recognised the complex interaction between attention, memory, information processing and how these affect learning. The selected battery aimed to provide us with an overall baseline of some of the cognitive processes important for learning and wellbeing, but within the limitations imposed by the wider remit of the study. Note This was the first UK study with a comparison group to collect a wide range of data from long term survivors of perinatally acquired HIV infection. 

The decision of combining some of the included tests was based more on functions (e.g. immediate or delayed recall) than specific underlying neurocognitive concepts. However, acknowledging the relevance of the points made by the reviewer, we have included separate scores of each of the tests on the S1 table. The table would provide the reader with the opportunity of exploring performance of participants on each test within each study group. 

We grouped the tests into domains based on the rationale alluded above, and used Cronbach α (a measure of internal consistency) to check these assumptions. We have clarified this in the methods section.

4. Although the results for the Verbal Learning PHIV C vs PHIV-C are statistically different, the confidence intervals are completely overlapping. Moreover, the z-scores are very similar (1.3 vs 1.5). This translates to a standard score of 78 vs 81, respectively. Is this really clinically meaningful? These scores may result in very similar behavioral presentation. Effect sizes should be reported to demonstrate magnitude of effects.

Response: The results of the ANOVA analysis show that the average z-scores in the three HIV/CDC groups are statistically significantly different which is also found in the regression analysis where HIV/CDC group was a statistically significant predictor of verbal learning scores. In both analyses PHIV/C had the lowest or predicted the lowest scores, followed by PHIV/no C and then the HIV- group. We have amended the wording of our interpretation of the results to suggest a trend can be seen. We have also presented the test scores by HIV/CDC group in S1 Table and the results of the regression analysis in S2 Table.

5. More clarity is warranted with regard to the aim and methods of the regression analysis:

a. What is the goal of the regression analysis? If part of the primary objective of the paper is to determine what factors might predict poorer cognitive functioning in this group, that should be made explicit in the introduction/aims.

Response: This has been added to the introduction.

b. IQ/education are important predictors of cognitive functioning. Were these controlled for in the regression to determine what factors predict poor outcomes? If not, why?

Response: IQ was not collected as part of the AALPHI study. Highest qualification was collected but was poorly completed with lots of missing data and therefore was not included in the analysis.

c. The authors note that the predictors used in the regression can be found in Table 1. Table 1 has over 20 variables. Are all of these included in the regression? I would think the analysis would be underpowered for that many variables. Please clarify and provide justification for the predictors.

Response: The variables with a * asterisk in Table 1 were included as predictors in the overall regression analysis. We have clarified this in the methods section. The variables chosen were reviewed by the AALPHI Steering Committee and included important variables which have been highlighted in other studies. 

d. An additional analysis with only PHIV was conducted. What is the rationale for this? Was this decided a priori or is this a sensitivity analysis? This needed clarification.

Response: This analysis was conducted to investigate the effect of HIV-related factors on cognitive performance and was decided a priori. This has been clarified in the introduction and methods section.

6. Raw score means, SDs, and ranges for all cognitive testing should be presented. This is necessary for interpretation of the results.

Response: These data have now been included in a Supporting Information file (S1 Table).

Minor Revisions by Section

Abstract

1. The title and Abstract background make it sound like only learning and memory was assessed. Consider updating both.

Response: We thank the reviewer for their comment. As the main finding is related to these domains, we think the focus of the title is appropriate. The methods section of the abstract confirms the wider assessments conducted. 

2. Acronyms are not defined before use. (e.g., CDC C diagnosis, ART)

Response: Thank you for your comment. They are now defined.

3. The original aim is to compare PHIV to HIV-PV, but there is a three-group comparison reported in results.

Response: We have amended the abstract and introduction to better reflect the three groups compared in the analysis.

4. Results of the linear regressions are not mentioned.

Response: We have included the regression analysis in the abstract and added it as an aim in the introduction.

5. The conclusion made in the abstract that results suggest earlier ART leads to better outcome is not support by the results reported in the abstract. There is no mention of ART in the result section.

Response: The reviewer is right in their observation. Our analysis did not show an association with ART exposure or age of ART initiation. However, our analysis clearly demonstrated previous AIDS-defining conditions differentiated the level of cognitive impairment in our study population. Since the main aim of ART is restoring or preventing deterioration of immune function and hence, preventing opportunistic infections or other AIDS-defining conditions; AIDS-defining conditions can be considered as a surrogate marker for delayed ART initiation which may or may not be due to late diagnosis with HIV infection. Participants with AIDS-defining conditions were mainly started on ART when presenting with these complications. Those participants who were started on ART before their immune functions was severely deteriorated showed better performance in all neuropsychological tests and, taking these two points together, we believe the conclusion is well supported. We have clarified our conclusions in the abstract and discussion.

Introduction

6. The connection to education in the introduction is a little unclear. This might be an important implication of the work; however, it is not directly related to the study aim.

Response: We decided to focus on executive function, verbal learning, and memory because these may impact outcomes in education and employability. We have amended the introduction to reflect this. 

7. A rationale should be provided for using the HIV-negative control group. I assume this control group accounts for environmental and psychosocial factors related to HIV exposure. This should be made explicit.

Response: This rationale has been added to the methods section, where we think it fits better.

8. In line with the above, there is clearly good reason for comparing groups with and without a CDC C diagnosis separately, but the rationale should be made explicit.

Response: We have amended the introduction to making the inclusion of all three groups clearer.

9. Why are no a prioiri hypotheses reported?

Response: We have clarified the aims of the analysis in the introduction.

Method

10. The first sentence needs editing. It reads like the study was evaluating the impact of HIV infection and ART on HIV- young people as it is currently written.

Response: This has been amended.

11. Please report any exclusion criteria for the larger study.

Response: This has been added.

12. A lot of space is used to discuss the consent/assent process. This is good/ethical practice, but unnecessarily detailed for a manuscript.

Response: Some of the detail has been removed from this section.

13. Cognitive testing described to have occurred during a face-to-face interview. It may be better described as an interview with cognitive testing.

Response: This has been reworded. 

14. Please define the CDC Clinical classification class C past history of an AIDS defining illness.

Response: The CDC C classification includes over 20 conditions. We have referenced the paper listing the conditions (apologies an incorrect reference was used in the original manuscript and has now been updated). If the editor would like all the conditions listed in the manuscript, then we can add them. 

Statistical analysis

15. Specify what comparisons are being made using chi square tests/Wilcoxon/KW.

Response: This has been clarified.

Results

16. Were there any significant differences between your groups on important demographic or clinical factors such as education, SES, psychiatric diagnoses? I believe these comparisons are made in Table 1, but the authors should include a statement in the results and then point readers to the table.

Response: A sentence describing the significant differences has been added to the results section and Table 1 referenced. 

17. Information on variable distribution/assumptions underlying statistical analyses should be reported.

Response: The normality of the data was assessed using visual plots and Sharpio-Wilk tests. Fisher’s exact tests for proportions were used if less than 80% of the cells had an expected value of five or greater. This information has been added to the Methods section.

Discussion

18. It would be interesting to discuss in more detail why your control group also performed lower than the normative sample on several tests to contextualize the results. For example, what common factors between all three groups could have impacted cognitive performance on these measures and is this consistent with the literature?

Response: Young people recruited as part of the comparison group were mainly siblings of PHIV or had a parent living with HIV and therefore, have been affected by HIV since their early childhood, making them comparable to our PHIV participants as they were likely to be exposed to all unmeasured factors that can affect development of cognitive functions. For instance, the majority of youngsters in both groups will have had a mother who was HIV positive and parental ill health, both physical and mental, and death and loss is much higher than in general population. Inconsistency or changes in care and the emotional and economic consequences of living with chronic parental ill health as well as the anxiety and confusion and uncertainty associated with managing a secret and stigmatising family diagnosis and add to the obstacles faced by many of these youngsters. These early chronic factors affect later learning and psychological outcomes and may explain the finding that scores on many tests fell below the published test norms in all study groups. 

 We have added more details about the HIV- comparison group to the methods section.

19. Based on what is reported in the discussion, the HIV+ cohort in this study had impairments in cognitive domains that differed from previously published worked. This should be discussed.

Response: Our study reported performance on all cognitive function domains below the normative mean of each test, which is consistent with several previous studies. In the discussion, we speculated unmeasured factors, such as socioeconomic variables, might affect testing performance (we have added a recent study Mbewe et al JAIDS 2022) and stated the importance of including an appropriate comparison group (i.e. HIV exposed but uninfected). Our results are also consistent with previous studies showing poorer performance in PHIV with previous AIDS-defining conditions when compared to those with less severe HIV disease and exposed but uninfected young people. 

20. There is a note to add “blackmore reference” in the discussion.

Response: Thanks for pointing this out, it has been removed. The reference has already been included.

21. It is unclear why the second limitation is a limitation of the study. Was identifying differential effect of antiretroviral drugs on cognition a goal? Is there a reason to believe that different treatments affect cognition in different ways? If so, this should be made more explicit.

Response: Virological escape and compartmentalisation have been reported in PHIV. In patients treated with ART combinations with poorer penetration to the CNS or in those with other conditions affecting blood-brain barrier integrity, viral particles and consequent inflammatory mediators can be observed in CSF, even in patients with undetectable HIV-RNA in peripheral blood. Detectable HIV-RNA in CSF has been associated with a number of neurological negative outcomes, including impaired cognition. In addition, some anti-retroviral drugs may cause neuropsychiatric adverse effects some of which may interfere with neuropsychological testing. Therefore, not being able to explore the impact of ART exposure on cognition we believe is a limitation of the study. 

22. A limitation of this study is only using active recall to test memory (i.e., no recognition memory test from HVLT), therefore it is unclear if deficits in delayed memory extend to recognition. If this cohort has impaired learning and recall, but not encoding (i.e., recognition memory is average), this would suggest a frontal executive, as opposed to amnestic mesial temporal profile.

Response: We agree with the reviewer, not including the verbal recognition component of the revised HVLT limits our ability of getting a full interpretation of the results if the aim of the analysis was understanding the neurological pathway leading to the observed performance level. However, the testing battery also included the NIH Toolbox Picture Sequence Test to explore episodic memory which measures episodic memory. Episodic memory could inform on medial temporal lobe function. 

The neuropsychological test battery used for this analysis was one of several components included in the main study visit, which was decided in advance should not last more than two hours to make it more likely for each participant to complete the full assessment. That was the main reason for restricting the number of neuropsychological tests. In any case, the aim of the analysis conducted was to compare performance on specific cognitive function domains between PHIV and appropriately selected HIV negative young people.

23. The authors conclude that their findings support that earlier ART protects against cognitive development. Was age of ART included in the regression model? Did it predict outcomes? It seems you have the data to test this assumption, but it is not clearly reported anywhere in the manuscript.

Response: We agree with the reviewer (also see earlier response to minor revisions, comment 5), the tone of the conclusion seems to suggest ART initiation at an earlier age, may protect cognitive function. However, the intention was to state ART initiation at an earlier stage of the HIV disease (i.e. before it reaches severe immunosuppression and opportunistic diseases occur) would reduce the risk of cognitive impairment. The text of the manuscript was amended to reflect the basis for the conclusion. Age of ART initiation was included as a predictor in the PHIV only model, but it was not associated with cognitive performance. The methods section has been amended to clarify which variables were considered as predictors in the models. 

Tables

24. Table 1 is very large. I would recommend splitting up into more than one table.

Response: The table has been split into Table 1 (Characteristics of all participants) and Table 2 (HIV-related factors for PHIV group).

25. Should present results of the linear regressions in a table for transparency.

Response: The results of the linear regressions have been added as Supplementary Information (S2 and S3 Tables).

Reviewer #2: Abstract

1. Background paragraph of abstract emphasizes relevance of learning and memory for successful educational and career progression, but this was not examined in this study. Authors should consider a more relevant background justification that would more directly apply to this work and supporting analyses.

Response: See response to reviewer 1, minor revisions, comment 6.

2. “CDC C” diagnosis term in abstract would not be known to a general scientific audience. Consider briefly defining.

Response: The definition of CDC C has been added to the abstract.

Methods

1. The section on the consent process can be shortened/streamlined.

Response: Some of the detail has been removed from this section.

2. It would be helpful for the authors to briefly explain why it would be relevant to compare PHIV/no C and PHIV/C. This is done within the discussion, but it would be beneficial for the reader to understand the reasoning for this comparison from the outset (perhaps within the introduction?)

Response: Our group and others have reported advanced HIV disease, defined as CDC stage C, to be an important risk factor for poorer cognitive function. In the same study population during their first neuropsychological assessment, participants in the PHIV/C performed worse than PHIV/no C and HIV- young people in some specific function domains, including executive function, speed of information processing and fine motor skills. Of note, at the initial assessment, we observed no difference between PHIV/no C and HIV- young people in any cognitive domain testing performance (Judd et al Clin Infect Dis 2016). We do agree with the reviewer, the rationale for comparing participants with different CDC status could be better explained on the text and we have amended the introduction accordingly. 

3. Executive functioning and memory performance can be directly impacted by attention, and attention difficulties are documented in HIV. Was there a reason attention measures were not included in this battery?

Response: We do agree with the reviewer on the impact attention impairment would have on executive functioning and memory. Due to time constrains we were unable to include a more comprehensive neuropsychological testing battery and therefore, we decided to include tests that could explore additional cognitive function domains. For instance, as part of the tests used to explore executive functioning in our participants, we included the NIH Toolbox Flanker Inhibitory Control and Attention test that directly explores attention. So, if one wanted to explore the impact of attention impairment on the overall assessment of executive functioning, one could look into the Flanker Inhibitory Control test score. However, the aim of the analysis conducted was to compare performance on specific cognitive function domains, including executive functioning, between PHIV and appropriately selected HIV negative young people.

Results

1. While it is logical that multiple test scores within a domain would be averaged to present results in a cohesive manner, the authors should make explicit that the scores were sufficiently statistically similar for this to be appropriate. Please include a table summarizing mean performance per group for all individual tests using a common metric (e.g., z scores). Also explain the criteria for averaging across tests within a domain; for example, did the authors ensure that mean test scores within a domain were not statistically different from each other? I imagine that there may have been some variability in performance for within-domain tests. Is it possible that potential effects were attenuated because of variable performance within domains?

Response: Mean test scores and z-scores for each test have now been included in a Supporting Information file (S1 Table). As stated before, we grouped the tests into domains based on theoretical assumptions and used Cronbach α (a measure of internal consistency) to check these assumptions. We have clarified this in the methods section.

Discussion

1. There were some lower cognitive scores amongst participants of Black race found in adjusted multivariable models. It is important that the authors address potential explanations for this within the discussion.

Response: This follows a trend reported in our previous paper and we outline potential reasons for this trend in our previous paper. We have added a summary comment to the discussion of this paper.

General comment for consideration:

1. The term “Black” generally refers to a race/racial category, as you can have participants who are “Black” but report different ethnic backgrounds. The authors can consider these recently published suggestions for reporting of race or ethnicity: https://jamanetwork.com/journals/jama/fullarticle/2776936

Response: We have added to the methods section that ethnicity was self-reported, and clarified the categories used and which categories were combined for the purpose of the analysis. We have also capitalised the names of ethnicities. We have not reported detailed ethnicity breakdowns due to small numbers in some categories.

---

## [Decision Letter · Decision Letter 1]

27 Apr 2022

PONE-D-21-16526R1

Learning and memory function in young people with and without perinatal HIV in England

PLOS ONE

Dear Dr. Arenas-Pinto,

Thank you for submitting your manuscript to PLOS ONE. After careful consideration, we have decided that your manuscript does not meet our criteria for publication and must therefore be rejected.

Specifically:

Your paper went out to one reviewer who assessed the revisions against the original reviews.  The reviewer noted that you had not satisfactorily addressed the initial concerns around situating the study in the literature, differentiating between this study and the previously published version of this study and that your grouping of tests required more justification. Given that the concerns are the same as the last time despite having a chance to address them I am afraid that the changes are just too onerous and I will have to reject the paper.

I am sure this news is not what you wanted to hear but I do not believe it would be fruitful to go back and forth on the same issues. I am sorry that we cannot be more positive on this occasion, but hope that you appreciate the reasons for this decision.

Kind regards,

Lori Buchanan, Ph.D.

Academic Editor

PLOS ONE

Reviewers' comments:

Reviewer's Responses to Questions

**Comments to the Author**

1. If the authors have adequately addressed your comments raised in a previous round of review and you feel that this manuscript is now acceptable for publication, you may indicate that here to bypass the “Comments to the Author” section, enter your conflict of interest statement in the “Confidential to Editor” section, and submit your "Accept" recommendation.

Reviewer #1: (No Response)

2. Is the manuscript technically sound, and do the data support the conclusions?

Reviewer #1: Partly

3. Has the statistical analysis been performed appropriately and rigorously? 

Reviewer #1: No

4. Have the authors made all data underlying the findings in their manuscript fully available?

Reviewer #1: Yes

5. Is the manuscript presented in an intelligible fashion and written in standard English?

Reviewer #1: Yes

6. Review Comments to the Author

Reviewer #1: PONE-D-21-16526R1

The authors appropriately addressed many of the minor concerns and clarifications recommended by the reviewers; however, there are a few major concerns that were not addressed in this revision.

1. A more thorough review of the background literature would be helpful.

2. It is still not clear how this study adds to the previously published study. It sounds like the previous study included the same three groups and a variety of cognitive measures (presumably EF, learning, and memory measures). Is this a replication?

3. I still have concerns about the combination of neuropsychological tests scores. The authors have grouped the tests “based on theoretical assumptions”, but there is empirical evidence to suggest different groupings would make more sense based on the underlying neurocognitive functions assessed by these measures. The rationale for the groups needs to be strengthened.

4. The authors did not address the concern regarding the difference between statistical and clinical significance. Adding effect sizes would help readers understand the magnitude of effect for the statistically significant group difference.

7. PLOS authors have the option to publish the peer review history of their article (what does this mean?). If published, this will include your full peer review and any attached files.

Reviewer #1: No

- - - - -

---

## [Author Response · Author response to Decision Letter 1]

10 Jun 2022

Appeal request form - Response to Reviewer #1: PONE-D-21-16526R1

We provide a point-by-point response to the reviewer here. The reviewer’s concerns are in bold text and our response in italics.

As a way of background, in December 2021 we received numerous comments from two reviewers as well as a summary from the Academic Editor, in which the Editor noted that

“The reviewers and I agree that this work is of importance and that it should be published but both reviewers have highlighted areas where the paper can be strengthened to have a greater impact.”

This implies that the manuscript was already publishable but that improvements could lead to a stronger paper. We responded robustly to all of the comments, and made significant changes to the manuscript, although in hindsight could have improved the introduction further.

The authors appropriately addressed many of the minor concerns and clarifications recommended by the reviewers; however, there are a few major concerns that were not addressed in this revision.

1. A more thorough review of the background literature would be helpful.

In our last revision, we added more details to the introduction about our previous study looking at cognitive function at interview 1.

We have now added references and text to contextualise our study within recent literature. See tracked changes attached.

2. It is still not clear how this study adds to the previously published study. It sounds like the previous study included the same three groups and a variety of cognitive measures (presumably EF, learning, and memory measures). Is this a replication?

We actually wanted to measure for a second time performance in some of the cognitive functions we explored in interview 1, which were most affected in interview 1, but we decided not to look at change over time on any of these domains because we were using different tests. We decided to use different tests to increase comparability of our cohort to a similar study in the USA. 

In our previous response to reviewer comments we said:

“As the reviewer mentioned, we have explored cognitive function performance in the AALPHI cohort at baseline. In the initial assessment we found no overall difference in neuropsychological testing performance between young people living with HIV, who have never developed any AIDS-defining complication, and HIV-negative young people. However, performance on tests exploring memory and information processing cognitive domains seem to be poorer in PHIV, particularly those who have experienced AIDS-defining complications in the past. Therefore, the round of neuropsychological assessment we are presenting in this manuscript, focused predominately on learning and memory cognitive function domains because of their implications on educational and employment outcomes. We have amended the introduction to reflect this.”

As stated above, we have already reported some differences in cognitive testing performance between participants with previous AIDS-defining conditions compared to those living with HIV but that have never progressed to CDC stage C conditions. The aim of this second neuropsychological assessment was to further explore cognitive domains considered particularly relevant to school progression and transition to adulthood. The assessment conducted is NOT a replication of our previous analysis. 

We therefore have made no further changes.

3. I still have concerns about the combination of neuropsychological tests scores. The authors have grouped the tests “based on theoretical assumptions”, but there is empirical evidence to suggest different groupings would make more sense based on the underlying neurocognitive functions assessed by these measures. The rationale for the groups needs to be strengthened.

We believe that the reviewer has strong feelings against grouping tests in a way different to what they think is appropriate. We do not think it should be up to the reviewer to decide what rationale for combining tests is valid or not, but despite our explanations on the matter, they still seem unhappy.

To clarify, we used more than one test to explore three of the cognitive function domains evaluated. When exploring Executive Functioning, we used two NIH ToolBox tests that are designed and validated for that use (i.e. flanker inhibitory control and attention test and the dimensional change card sort test. The latest one, specifically designed and validated to explore cognitive flexibility). 

When exploring language-related functions, we used the Weschler Individual Achievement Test® and the British Picture Vocabulary Scale, both commonly used and validated tests for this purpose. 

Finally, because focusing on memory and learning skills was our main aim, we did conduct a number of different tests exploring that function domain. However, precisely because we wanted to investigate performance on each one of the components or interrelated specific tasks, we reported separately scores on verbal learning, verbal recall (HVLT-R) and episodic memory (NIH ToolBox: list sorting and picture sequence test). We also reported performance on attention and information processing speed separately. 

Therefore, we seriously struggle to understand the objections the reviewer may have on our approach to report performance by cognitive function domain. We have however amended the methods section to reflect the rationale for grouping neuropsychological tests (lines 125 and 126 of the amended draft). 

We previously responded to this comment with the following:

“We do agree that the findings do not clarify the specific neurocognitive processes most affected by the presence of HIV in the developing brain. The choice of tests recognised the complex interaction between attention, memory, information processing and how these affect learning. The selected battery aimed to provide us with an overall baseline of some of the cognitive processes important for learning and wellbeing, but within the limitations imposed by the wider remit of the study. Note This was the first UK study with a comparison group to collect a wide range of data from long term survivors of perinatally acquired HIV infection. The decision of combining some of the included tests was based more on functions (e.g. immediate or delayed recall) than specific underlying neurocognitive concepts. However, acknowledging the relevance of the points made by the reviewer, we have included separate scores of each of the tests on the S1 table. The table would provide the reader with the opportunity of exploring performance of participants on each test within each study group. We grouped the tests into domains based on the rationale alluded above, and used Cronbach α (a measure of internal consistency) to check these assumptions. We have clarified this in the methods section.”

4. The authors did not address the concern regarding the difference between statistical and clinical significance. Adding effect sizes would help readers understand the magnitude of effect for the statistically significant group difference.

We did not see any concerns regarding the difference between statistical and clinical significance in the previous set of reviewer comments. We responded to the request to add effect sizes in our previous response. We added further information on significant differences, and we added a supplementary table showing separate scores of each test. The discussion of the results does not include any speculation on the clinical impact of the differences we observed between the study groups on specific neuropsychological testing scores. Rather, we provided the reader with the scores by test and cognitive functional domain. Exploring the clinical impact of the differences observed was beyond the scope of this analysis and therefore, we have not made any further changes here.

---

## [Decision Letter · Decision Letter 2]

28 Jul 2022

PONE-D-21-16526R2Learning and memory function in young people with and without perinatal HIV in EnglandPLOS ONE

Dear Dr. Arenas-Pinto,

Thank you for submitting your manuscript to PLOS ONE. After careful consideration, we feel that it has merit but does not fully meet PLOS ONE’s publication criteria as it currently stands. Therefore, we invite you to submit a revised version of the manuscript that addresses the points raised during the review process.

ACADEMIC EDITOR: Please insert comments here and delete this placeholder text when finished. Be sure to:Indicate which changes you require for acceptance versus which changes you recommendAddress any conflicts between the reviews so that it's clear which advice the authors should followProvide specific feedback from your evaluation of the manuscriptPlease ensure that your decision is justified on PLOS ONE’s publication criteria and not, for example, on novelty or perceived impact.

We look forward to receiving your revised manuscript.

Kind regards,

Lori Buchanan, Ph.D.

Academic Editor

PLOS ONE

Journal Requirements:

Additional Editor Comments (if provided): **Please make sure you read the reviewer's comments. In particular please add clarity to the difference between this and your previous study (pt 2).  The addition of standardized scoring seems like something that could easily be done and like the reviewer I am puzzled about why that isn't included. Please add those or provide an explanation about why you are resisting that suggestion.**

Reviewers' comments:

Reviewer's Responses to Questions

**Comments to the Author**

1. If the authors have adequately addressed your comments raised in a previous round of review and you feel that this manuscript is now acceptable for publication, you may indicate that here to bypass the “Comments to the Author” section, enter your conflict of interest statement in the “Confidential to Editor” section, and submit your "Accept" recommendation.

Reviewer #1: (No Response)

2. Is the manuscript technically sound, and do the data support the conclusions?

Reviewer #1: Yes

3. Has the statistical analysis been performed appropriately and rigorously? 

Reviewer #1: Yes

4. Have the authors made all data underlying the findings in their manuscript fully available?

Reviewer #1: Yes

5. Is the manuscript presented in an intelligible fashion and written in standard English?

Reviewer #1: Yes

6. Review Comments to the Author

Reviewer #1: Please see the attached file.

PONE-D-21-16526R2

1. The introduction and background review has been improved.

2. I am not doubting that the study adds to the literature. My confusion on what this study adds to their previous work comes from the lack of specificity in Line 70-77 (bolded).

“We have previously reported similar performance in neuropsychological testing in young people living with HIV, without a prior history of AIDS-defining complications, and a comparable group of HIV negative (HIV-) young people (YP) within the Adolescents and Adults Living with Perinatal HIV (AALPHI) cohort study (7). Of note, PHIV with a past history of AIDS74

defining conditions showed poorer performance in all cognitive function domains explored. In this paper we report on further cognitive testing performance of young people with PHIV

(with or without a previous AIDS-defining illness) and a comparable 76 group of HIV- young people.”

The difference between their first and second study is made clearer in their response to reviewers, but the rationale has still not been added to the introduction for readers. If I am interpreting correctly, the authors were (1) further clarifying the differences in memory and information processing found in previous work and (2) used tests more widely used in the literature to compare to other findings. If I am interpreting this correctly, the authors could add this rationale into the manuscript to increase clarity for readers.

3. There is no one valid way to combine neuropsychological test scores. I provided empirical literature regarding the underlying neurocognitive functions that those tests measure. I did not state the authors must group tests as I suggested, only that it should be considered. For example, “verbal delayed recall” and “memory” tests are reported separately, but verbal delayed recall is memory. The authors show that the mean and Z-scores for the combined tests are comparable, which is helpful. Perhaps changing the name of the categories to better reflect the function they are attempting to assess would help.

4. Including the raw cognitive test data was very helpful. However, adding standardized effect sizes along with statistical significance testing demonstrates the magnitude of group difference and is common practice.

https://www.ncbi.nlm.nih.gov/pmc/articles/PMC3444174/#:~:text=Effect%20size%20helps%20readers%20understand,the%20Abstract%20and%20Results%20sections.

7. PLOS authors have the option to publish the peer review history of their article (what does this mean?). If published, this will include your full peer review and any attached files.

Reviewer #1: No

---

## [Author Response · Author response to Decision Letter 2]

11 Aug 2022

1. The introduction and background review has been improved. 

We thank the reviewer for their comment.

2. I am not doubting that the study adds to the literature. My confusion on what this study adds to their previous work comes from the lack of specificity in Line 70-77 (bolded). 

 “We have previously reported similar performance in neuropsychological testing in young people living with HIV, without a prior history of AIDS-defining complications, and a comparable group of HIV negative (HIV-) young people (YP) within the Adolescents and Adults Living with Perinatal HIV (AALPHI) cohort study (7). Of note, PHIV with a past history of AIDS74

defining conditions showed poorer performance in all cognitive function domains explored. In this paper we report on further cognitive testing performance of young people with PHIV

(with or without a previous AIDS-defining illness) and a comparable 76 group of HIV- young people.”

The difference between their first and second study is made clearer in their response to reviewers, but the rationale has still not been added to the introduction for readers. If I am interpreting correctly, the authors were (1) further clarifying the differences in memory and information processing found in previous work and (2) used tests more widely used in the literature to compare to other findings. If I am interpreting this correctly, the authors could add this rationale into the manuscript to increase clarity for readers. 

We thank the reviewer for their suggestion. We have amended the introduction to address the point.

3. There is no one valid way to combine neuropsychological test scores. I provided empirical literature regarding the underlying neurocognitive functions that those tests measure. I did not state the authors must group tests as I suggested, only that it should be considered. For example, “verbal delayed recall” and “memory” tests are reported separately, but verbal delayed recall is memory. The authors show that the mean and Z-scores for the combined tests are comparable, which is helpful. Perhaps changing the name of the categories to better reflect the function they are attempting to assess would help.

We agree with the reviewer’s point. Verbal learning and verbal delayed recall are functions related to the memory domain. However, we consider cognitive functions related to management of verbal language particularly relevant to school progression and employability. Therefore, in addition to including tests to explore verbal application, we also considered important to provide the readers with the opportunity of looking at memory functions related to the used of verbal language. 

We provide the scores of each individual test and hence, the reader has the possibility of observing performance on every function domain and every test explored. 

4. Including the raw cognitive test data was very helpful. However, adding standardized effect sizes along with statistical significance testing demonstrates the magnitude of group difference and is common practice. 

https://www.ncbi.nlm.nih.gov/pmc/articles/PMC3444174/#:~:text=Effect%20size%20helps%20readers%20understand,the%20Abstract%20and%20Results%20sections.

We agree with the reviewer and have added standardized effect sizes to table S1; please see additional rows entitled “standardised score” for each domain/ test.

---

## [Editor Report · Decision Letter 3]

15 Aug 2022

Learning and memory function in young people with and without perinatal HIV in England

PONE-D-21-16526R3

Dear Dr. Arenas-Pinto,

We’re pleased to inform you that your manuscript has been judged scientifically suitable for publication and will be formally accepted for publication once it meets all outstanding technical requirements.

Kind regards,

Lori Buchanan, Ph.D.

Academic Editor

PLOS ONE

Additional Editor Comments (optional):

thanks for your careful and responsive treatment of reviewers' comments.
---

## [Editor Report · Acceptance letter]

6 Sep 2022

PONE-D-21-16526R3 

Learning and memory function in young people with and without perinatal HIV in England 

Dear Dr. Arenas-Pinto:

I'm pleased to inform you that your manuscript has been deemed suitable for publication in PLOS ONE. Congratulations! Your manuscript is now with our production department. 

Kind regards, 

on behalf of

Dr. Lori Buchanan 

Academic Editor

PLOS ONE